# NFκB-Mediated Mechanisms Drive PEDF Expression and Function in Pre- and Post-Menopausal Oestrogen Levels in Breast Cancer

**DOI:** 10.3390/ijms232415641

**Published:** 2022-12-09

**Authors:** Naomi Brook, Jespal Gill, Arun Dharmarajan, Arlene Chan, Crispin R. Dass

**Affiliations:** 1Curtin Medical School, Curtin University, Bentley, WA 6102, Australia; 2Curtin Health Innovation Research Institute, Bentley, WA 6102, Australia; 3Pathwest, Fiona Stanley Hospital, Murdoch, WA 6150, Australia; 4Department of Biomedical Sciences, Faculty of Biomedical Sciences and Technology, Sri Ramachandra Institute of Higher Education and Research, Chennai 600116, India; 5Breast Cancer Research Centre-Western Australia, Hollywood Private Hospital, Nedlands, WA 6009, Australia

**Keywords:** pigment epithelium-derived factor, breast cancer, metastasis, oestradiol, estrone, estrone sulphate, menopausal status

## Abstract

Pigment epithelium-derived factor (PEDF) protein regulates normal bone, with anti-tumour roles in bone and breast cancer (BC). Pre- and post-menopausal oestrogen levels may regulate PEDF expression and function in BC, though the mechanisms behind this remain unknown. In this study, in vitro models simulating pre- and post-menopausal bone microenvironments were used to evaluate if PEDF regulates pro-metastatic biomarker expression and downstream functional effects on BC cells. PEDF treatment reduced phosphorylated-nuclear factor-κB p65 subunit (p-NFκB-p65), tumour necrosis factor-α (TNFα), C-X-C chemokine receptor type-4 (CXCR4), and urokinase plasminogen activator receptor (uPAR) in oestrogen receptor (ER)+/human epidermal growth factor receptor-2 (HER2)- BC cells under post-menopausal oestrogen conditions. In triple negative BC (TNBC) cells, PEDF treatment reduced pNFκB-p65 and uPAR expression under pre-menopausal oestrogen conditions. A potential reciprocal regulatory axis between p-NFκB-65 and PEDF in BC was identified, which was BC subtype-specific and differentially regulated by menopausal oestrogen conditions. The effects of PEDF treatment and NFκB inhibition on BC cell function under menopausal conditions were also compared. PEDF treatment exhibited superior anti-viability effects, while combined PEDF and NFκB-p65 inhibitor treatment was superior in reducing BC cell colony formation in a subtype-specific manner. Lastly, immunohistochemical evaluation of p-NFκB-p65 and PEDF expression in human BC and bone metastases specimens revealed an inverse correlation between nuclear PEDF and NFκB expression in bone metastases. We propose that menopausal status is associated with a PEDF/NFκB reciprocal regulatory axis, which drives PEDF expression and anti-metastatic function in a subtype-specific manner. Altogether, our findings identify pre-menopausal TNBC and post-menopausal ER+/HER2- BC patients as target populations for future PEDF research.

## 1. Introduction

Breast cancer (BC) is the most common cancer affecting women [1], with BC risk increased by younger age at menarche, older age at menopause, and in pre-menopausal woman compared to post-menopausal women of the same age [2]. Bone is a very common site of metastatic involvement in patients with BC [3]. Oestrogen protects the pre-menopausal bone environment against osteoclast-mediated bone loss [4]. Circulating oestrogen levels change with menopause which can enhance pathological bone loss and BC bone metastasis post-menopause [5]. Anti-resorptive agents such as bisphosphonates, along with chemotherapy can increase both overall survival and disease-free survival (DFS) in post-menopausal adjuvant BC patients [6]. Current bone-targeting therapies can improve survival in the adjuvant setting, and when given in combination with other systemic agents can reduce symptoms associated with bone metastases in metastatic BC patients, however, bone metastases remain incurable and new therapeutic strategies are still required.

Pigment epithelium-derived factor (PEDF) is a protein encoded by the serpin family F member 1 (SERPINF1/early population doubling level cDNA-1 (EPC1)) gene on chromosome 17p13.1 [7], expressed throughout the human body [8]. PEDF is an integral component of bone, where it interacts with key components of the bone extracellular matrix [9], including collagen [10]. PEDF regulates normal bone formation by regulating osteoclasts [11] and osteoblastogenesis [12], and also plays an anti-tumour role in osteosarcoma [13]. PEDF is also a potent anti-angiogenic factor and in concert with vascular endothelial growth factor (VEGF), a potent pro-angiogenic factor, is critically regulated by changing oestrogen levels to maintain normal endometrial tissue and vasculature growth during female ovulatory cycles [14]. Negative regulation of PEDF by oestrogen has also been reported in a number of other cell types and tissues [15]. Downregulation of PEDF in primary and secondary BC has been previously reported [16,17], and appears to be associated with increased tumour vascularisation and poor prognosis [18].

PEDF is a potential bone-targeting therapeutic candidate due to the role it plays in normal bone growth and its anti-tumour function observed in both breast and bone cancer. We have previously shown that nuclear PEDF expression in BC bone metastases is reduced in post-menopausal BC patients [19]. Furthermore, we also found reduced PEDF expression in the presence of post-menopausal oestrogen levels in BC cells in vitro, with PEDF anti-metastatic function enhanced under pre-menopausal conditions for TNBC cells, and under post-menopausal conditions for ER+/HER2- BC cells (manuscript submitted). We hypothesised that PEDF would differentially regulate the expression of pro-metastatic biomarkers under pre- versus post-menopausal oestrogen conditions in ER+/HER2- and TNBC cells. The aim of this study was to investigate the potential mechanisms driving differential PEDF expression and function in pre- versus post-menopausal oestrogen conditions in metastatic BC. We report here for the first time the effect of PEDF on the expression of key pro-metastatic biomarkers and associated downstream functional impact in 2 clinically relevant BC cell lines in the presence of physiologically relevant oestrogen levels reported in pre- and post-menopausal BC patients.

## 2. Results

### PEDF Differentially Regulates Biomarker Expression in Menopausal Conditions in ER + BC and TNBC Cells

We have previously shown PEDF expression is reduced in the post-menopausal setting in BC bone metastases [19]. Additionally, we found PEDF is downregulated under post-menopausal oestrogen conditions in ER+/HER2- BC and TNBC cells in vitro, and PEDF anti-metastatic function is enhanced under post-menopausal conditions for ER + BC cells and under pre-menopausal conditions for TNBC cells (manuscript submitted). As such, we were interested in the potential mechanisms driving differential PEDF expression and function under pre- versus post-menopausal oestrogen conditions, and specifically the pro-metastatic biomarkers potentially differentially regulated by PEDF under menopausal conditions in BC cells. Using immunocytochemistry (ICC) techniques, we investigated if PEDF regulates pro-metastatic biomarker expression under post-menopausal conditions in MCF-7, and under pre-menopausal conditions in MDA-MB-231 cells. Biomarkers selected for in vitro metastasis studies based on reported pro-metastatic function in BC bone metastasis are summarised in Table 1.

We found PEDF decreased phosphorylated-nuclear factor kappa B-p65 subunit (p-NFκB-p65) (*p* < 0.0001) (Figure 1I–L,c) and C-X-C chemokine receptor type 4 (CXCR4) (*p* = 0.0037) (Figure 1A–D,a) expression in post-menopausal conditions only compared to controls in ER+ MCF-7 cells. PEDF also downregulated CXCR4 expression compared to control in post-menopausal oestrogen conditions only in MDA-MB-231 cells (*p* = 0.0358) (Figure 1E–H,a). Interestingly, PEDF increased p-NFκB-p65 under pre-menopausal conditions in MCF-7 (*p* = 0.0061) (Figure 1I–L,c) and under post-menopausal conditions in MDA-MB-231 cells (*p* < 0.0001) (Figure 1M–P,c). PEDF also slightly increased CXCR4 expression under pre-menopausal conditions in both cell lines, although this did not reach statistical significance (Figure 1a). PEDF decreased tumour necrosis factor alpha (TNF-α) (Figure 1Q–T,e), urokinase-type plasminogen activator receptor (uPAR) (Figure 1Y–AB,g), membrane-type 1 matrix metalloproteinase (MT1-MMP) (Appendix A), and receptor activator of nuclear factor kappa-Β ligand (RANKL) (Appendix A) expression under both pre-menopausal (TNF-α: *p* = 0.0027; uPAR: *p* = 0.0145; MT1-MMP: *p* = 0.0003; RANKL: *p* = 0.0084) and post-menopausal (TNF-α: *p* = 0.0009; uPAR: *p* < 0.0001; MT1-MMP: *p* = 0.0005; RANKL: *p* = 0.0044) conditions compared to controls in MCF-7 cells. TNF-α (*p* = 0.0009) and uPAR (*p* < 0.0001) were downregulated by PEDF compared to controls under pre-menopausal conditions only (Figure 1U–X,AC–AF,e,g); while, MT1-MMP (*p* < 0.0001) and RANKL (*p* < 0.0001) were downregulated in post-menopausal conditions only in MDA-MB-231 cells (Appendix A). PEDF decreased urokinase-type plasminogen activator (uPA) expression under both pre-menopausal (*p* = 0.0003) and post-menopausal (*p* = 0.0106) conditions compared to controls in MDA-MB-231; whereas uPA was only significantly decreased under pre-menopausal conditions in MCF-7 (*p* = 0.0044) (Appendix A).

When comparing the effects of PEDF treatment on downregulating biomarker expression in pre- versus post-menopausal conditions, the inhibitory effect of PEDF on p-NFκB-p65 expression was greatest under post-menopausal conditions in MCF-7 and under pre-menopausal conditions in MDA-MB-231 cells (Figure 1d). PEDF downregulated TNF-α (*p* = 0.0027) and uPAR (*p* = 0.0002) to a greater extent in post-menopausal conditions, and RANKL (*p* = 0.0097) under pre-menopausal in MCF-7 (Figure 1f,h and Appendix A). PEDF had the greatest inhibitory effect on uPAR (*p* = 0.0027) expression under pre-menopausal conditions (Figure 1h), and on RANKL (*p* = 0.0002) and MT1-MMP (*p* < 0.0001) under post-menopausal conditions in MDA-MB-231 cells (Appendix A). No significant differences between PEDF-mediated uPA downregulation in pre- versus post-menopausal conditions were observed in either MCF-7 or MDA-MB-231 cells (Appendix A). These findings show that PEDF decreased pro-metastatic biomarker expression under post-menopausal conditions in ER+/HER2- BC cells; with PEDF-mediated p-NFκB-p65, TNFα, CXCR4, and uPAR downregulation specific to post-menopausal conditions. PEDF downregulated uPAR and TNF-α downregulation in pre-menopausal conditions in TNBC cells, with TNF-α decreased to a similar degree in both pre- and post-menopausal conditions. PEDF downregulated TNF-α, uPAR, and uPA under pre-menopausal conditions, and MT1-MMP, RANKL, uPA, and CXCR4 under post-menopausal conditions in TNBC cells. To summarise, PEDF exhibited the greatest downregulatory effect on p-NFκB-p65, TNF-α, CXCR4, and uPAR expression under post-menopausal conditions in ER+/HER2- BC, and on p-NFκB-p65 and uPAR under pre-menopausal conditions in TNBC cells.

NFκB emerged as a key pro-metastatic biomarker downregulated by PEDF in ER+/HER2- BC cells under post-menopausal conditions and in TNBC cells under pre-menopausal conditions. To investigate potential mechanisms driving enhanced PEDF anti-metastatic function in BC cells under menopausal conditions, we sought to determine if PEDF/NFκB reciprocal regulation exists in BC cells in vitro. Following treatment with a specific NFκB-p65 inhibitor (JSH-23), we measured PEDF expression levels in pre- versus post-menopausal oestrogens in MCF-7 and MDA-MB-231 cells. JSH-23 treatment increased PEDF expression compared to control under post-menopausal oestrogen conditions (*p* = 0.0002) (Figure 2C,D,a) but decreased PEDF under pre-menopausal oestrogen conditions in MCF-7 cells (*p* > 0.05) (Figure 2A,B,a), with the greatest effect seen in post-menopausal conditions (*p* = 0.0002, Figure 2b). In MDA-MB-231 cells, JSH-23 treatment increased PEDF expression compared to control under both pre-menopausal and post-menopausal oestrogen conditions (*p* < 0.0001, Figure 2E–H,a), with greater PEDF upregulation seen in pre-menopausal compared to post-menopausal oestrogen conditions (*p* < 0.0001, Figure 2b). These results indicate NFκB-p65 negatively regulates PEDF under post-menopausal conditions in ER+/HER2- BC and under pre-menopausal in TNBC cells. Our findings suggest a reciprocal regulatory axis between PEDF and NFκB-p65, which may be associated with post-menopausal circulating oestrogen levels in ER+/HER2- BC and pre- menopausal circulating oestrogens in TNBC cells.

As PEDF appears to be downregulated in post-menopausal ER+/HER2- BC bone metastasis [19] and NFκB-p65 regulation of PEDF in BC cells depends on menopausal oestrogen levels, we investigated the effects of PEDF and/or JSH-23 treatment on pro-metastatic biomarker expression under pre- and post-menopausal oestrogen conditions in MCF-7 and MDA-MB-231 cells (Figure 3). PEDF, JSH-23, and combined PEDF/JSH-23 treatment reduced p-NFκB-p65 expression compared to control under pre-menopausal (PEDF: *p* = 0.0002; JSH-23: *p = 0.0027*; PEDF/JSH-23: *p* = 0.0001) and post-menopausal (PEDF: *p* = 0.3437; JSH-23: *p* = 0.0007; PEDF/JSH-23: *p* = 0.0089) conditions in MDA-MB-231 (Figure 3I–P,a). However, only PEDF treatment reduced p-NFκB-p65 under post-menopausal conditions in MCF-7 (*p=* 0.0028; Figure 3A–H,a). The same treatments reduced TNFα expression under pre-menopausal (PEDF: *p* = 0.001; JSH-23: *p* = 0.7668; PEDF/JSH-23: *p* = 0.7593) and post-menopausal (PEDF: *p* = 0.2986; JSH-23: *p=* 0.0006; PEDF/JSH-23*: p <* 0.0001) conditions in MDA-MB-231 (Figure 3Y–AF,c). In MCF-7, all three treatments reduced TNFα under pre-menopausal conditions (PEDF: *p* = 0.0006; JSH-23: *p* = 0.0035; PEDF/JSH-23: *p* < 0.0001), however only PEDF treatment reduced TNFα compared to control under post-menopausal conditions (*p* = 0.0028; Figure 3Q–X,c). PEDF treatment reduced uPAR expression compared to controls under pre-menopausal conditions in MDA-MB-231 (*p* < 0.0001) (Figure 3BE–BL,e). In MCF-7, PEDF reduced uPAR under pre-menopausal (PEDF: *p* = 0.0023), and PEDF (*p* = 0.0014) and combined PEDF/JSH-23 (*p* = 0.014) reduced uPAR under post-menopausal conditions (Figure 3AW–BD, 3e). No significant trends were observed for any treatments on CXCR4 expression in MDA-MB-231 (Figure 3AO–AV,g). In MCF-7, JSH-23 reduced CXCR4 expression under pre-menopausal conditions (*p* = 0.0176), and both PEDF (*p* = 0.001) and JSH-23 (*p* = 0.0186) reduced CXCR4 under post-menopausal conditions (Figure 3AK–AN,g). Overall, PEDF treatment alone had the greatest effect on reducing metastatic biomarker expression under post-menopausal conditions in MCF-7 (p-NFκB-p65: *p* = 0.065, Figure 3b; TNFα: *p* = 0.002, Figure 3d; uPAR: *p* = 0.0149, Figure 3f; CXCR4: *p=* 0.0001, Figure 3h) and under pre-menopausal conditions in MDA-MB-231 (p-NFκB-p65: *p* = 0.0018, Figure 3b; TNFα: *p* < 0.0001, Figure 3d; uPAR: *p* = 0.0002, Figure 3f).

PEDF-mediated metastatic biomarker downregulation may be responsible for enhanced anti-metastatic action observed in pre- and post-menopausal oestrogen conditions in TNBC and ER+/HER2- BC cells, respectively. Therefore, we investigated if treatment with PEDF and/or specific NFκB-p65 inhibitor would affect anti-metastatic function in BC cells under menopausal oestrogen conditions in vitro (Figure 4). Colony formation assays showed that combined PEDF and JSH-23 treatment was most effective at reducing colony growth rate of MCF-7 under post-menopausal conditions, and JSH-23 treatment was more effective than PEDF treatment alone at reducing colony growth under the same conditions, reducing colony growth by ~50% (*p* = 0.0286, Figure 4G,H,O,P,R). In MDA-MB-231, all treatments prevented colony growth to a similar extent under pre-menopausal conditions, with a slight trend towards JSH-23 treatment alone having the greatest effect on reducing colony growth, however this did not reach statistical significance (Figure 4I,J,Q). No significant trends were observed for any effects of any treatment on colony growth in MCF-7 cells under pre-menopausal conditions, and in MDA-MB-231 cells under post-menopausal conditions.

Treatment with PEDF alone had the greatest ability to reduce viability of MCF-7 under post-menopausal conditions (*p* = 0.0161, Figure 4T), and MDA-MB-231 cells under pre-menopausal conditions (*p* = 0.001, Figure 4S) compared to either JSH-23 or combined PEDF/JSH-23 treatment. Altogether these results indicate that combined treatment with PEDF and an NFκB-p65 inhibitor reduced colony formation and therefore anchorage-independent growth of MCF-7 cells under post-menopausal oestrogen conditions, while these treatments had less impact on reducing colony growth of MDA-MB-231 cells under pre-menopausal conditions. PEDF treatment was most effective at reducing MCF-7 viability under post-menopausal oestrogen conditions and MDA-MB-231 viability under pre-menopausal oestrogen conditions.

As we previously characterised PEDF expression in samples of patient-matched primary BC and bone metastases tissue [19], we were interested in determining p-NFκB-p65 expression in a small subset of these specimens and correlating this with known PEDF scores. As such, we evaluated p-NFκB-p65 expression in a small subset of ER+/HER2- primary BC and bone metastases tissue (*n* = 4) and compared to PEDF expression levels previously determined in the same specimens from this patient cohort by our group [19], to determine if there is any association between PEDF and p-NFκB-p65 expression in human BC and bone metastases tissue (Figure 5). Nuclear p-NFκB-p65 expression was generally higher in BC cells in bone metastases compared to primary BC tissue (Figure 5B,D), whereas the same tissue samples previously stained for PEDF showed an opposite trend and nuclear PEDF expression was higher in BC cells in the primary tumour compared to BC cells in bone metastases tissue (Figure 5A,C, Table 2). Cytoplasmic PEDF was higher than p-NFκB-p65 in primary BC and bone metastases, while nuclear p-NFκB-p65 was higher than PEDF in both BC and bone metastases tissue (Figure 5, Table 2). This inverse correlation between nuclear PEDF and p-NFκB-p65 expression levels in BC cells reached statistical significance in bone metastases tissue (*p* = 0.0369, Table 2). To summarise, higher PEDF expression in primary BC tissue was associated with lower p-NFκB-p65 expression, while higher p-NFκB-p65 expression in bone metastases tissue was associated with lower PEDF expression.

## 3. Discussion

PEDF expression is regulated by sex hormones in normal and cancerous cells, with decreased PEDF expression in BC specimens and BC cells in vitro associated with post-menopausal status and post-menopausal oestrogen levels, respectively ([19], manuscript submitted). Furthermore, enhanced PEDF anti-metastatic function is associated with post-menopausal oestrogen levels in ER+/HER2- BC and pre-menopausal oestrogen levels in TNBC cells (manuscript submitted). However, the mechanisms driving differential PEDF expression and function under pre- versus post-menopausal oestrogen conditions in BC cells is unknown. Therefore, we investigated potential PEDF regulation of pro-metastatic biomarkers in ER+/HER2- BC and TNBC cells in the presence of physiologically relevant circulating oestrogen concentrations, as reported in pre- and post-menopausal BC patients [35]. PEDF exhibited the greatest downregulatory effect on p-NFκB-p65, TNF-α, CXCR4, and uPAR expression under post-menopausal oestrogen conditions in ER+/HER2- BC cells. In TNBC cells, PEDF had the greatest downregulatory effect on p-NFκB-p65 and uPAR expression under pre-menopausal conditions. To the best of our knowledge, this is the first report exploring potential links between PEDF and pro-metastatic CXCR4, MT1-MMP, RANKL, TNFα, uPA, and uPAR in BC metastasis. Although studies have previously shown an association between PEDF and NFκB in other cell types, only one previous study found an association between PEDF and NFκB expression in BC [36]. We have built on this information by investigating PEDF and p-NFκB-p65 in ER+/HER2- and TNBC cells under menopausal oestrogen conditions in vitro for the first time. Furthermore, this is the first study to evaluate PEDF and p-NFκB-p65 expression in human samples of BC and bone metastasis tissue. The biomarkers selected for investigation all play a role in BC bone metastasis and are regulated by oestrogens. For instance, overexpression of NFκB, a complex of transcription factors that regulate inflammatory and survival pathways [37], is associated with BC disease recurrence [24]. Furthermore, NFκB is negatively regulated by E2 and positively regulated by E1 in ER+ BC cells [38]. TNF-α is another pro-metastatic factor that increases BC cell growth, migration, and invasion [39], potentially by increasing MT1-MMP accumulation in membrane-associated lipid rafts [40]. Furthermore, E2 directly regulates TNF-α in ER+ BC cells [41]. CXCL12/CXCR4 signalling similarly regulates cell migration, proliferation, and survival pathways [42], with CXCR4 overexpression in primary BC is associated with high incidence of bone metastases by increasing BC cell homing to bone [20]. E2 also upregulates CXCL12 and CXCR4 expression in BC, which increases proliferation in vitro [21]. The serine protease uPA [43] and its receptor, uPAR [44], are pro-osteoclastic [45] and uPAR overexpression is an independent prognostic indicator of poorer survival in post-menopausal BC [46,47]. E2 regulates uPAR mobilisation from membrane-associated lipid rafts and promotes cancer cell migration [48]. Moreover, PEDF has previously been shown to play a regulatory role in the expression and function of NFκB, TNF-α, CXCR4, and uPAR. PEDF is an established anti-inflammatory protein that downregulates both NFκB and TNF-α under hypoxic conditions [49,50]. PEDF also downregulates CXCR4 and TNF, as well as increasing TNF-related apoptosis-inducing ligand to activate immunosurveillance in prostate cancer [51]. PEDF also interacts with uPA and uPAR by altering their localisation in osteosarcoma cells [52]. Interestingly, PEDF slightly upregulated CXCR4 expression under pre-menopausal oestrogen conditions in both cell lines. This finding may be of low physiological significance as it did not reach statistical significance, however this may be linked to reduced anti-metastatic PEDF function in MCF-7 cells under pre-menopausal conditions. Overall, we report for the first time that p-NFκB-p65, TNF-α, CXCR4, and uPAR are PEDF anti-metastatic targets specific to post-menopausal conditions in ER+/HER2- BC cells, with p-NFκB-p65 and uPAR also specific to pre-menopausal TNBC. Collectively, these results provide new insight into the role of PEDF in BC metastasis. Our novel studies suggest that PEDF-mediated downregulation of pro-metastatic mediators is associated with menopausal status and BC subtype, and likely accounts for the enhanced anti-metastatic action of PEDF observed in TNBC and ER+/HER2- BC cells under different menopausal oestrogen conditions (manuscript submitted).

Our study showed that inhibiting NFκB-p65 increased PEDF expression to a greater extent in pre-menopausal oestrogen conditions in TNBC and in post-menopausal oestrogen conditions in ER+/HER2- BC cells. This indicates that NFκB-p65-mediated downregulation of PEDF in BC may be dependent on molecular subtype and menopausal status. We also showed that PEDF downregulation of p-NFκB-p65 similarly depends on menopausal oestrogen levels in BC cells, suggesting for the first time the presence of a reciprocal regulatory relationship between PEDF and p-NFκB-p65 in BC, which may be associated with menopausal status. PEDF appears to protect against cell stress by negatively regulating NFκB in models of inflammation in a variety of cell types [53]. Likewise, pro-inflammatory activation of NFκB also regulates PEDF expression in adipocytes [54] and hepatocytes [55], indicating a reciprocal regulatory pathway between PEDF/NFκB in pro-inflammatory conditions. NFκB plays a central role in BC metastasis, and in combination with other pro-metastatic mediators, activates epithelial-mesenchymal transition [27], and promotes the expansion of stem-like ER+ BC cells and associated disseminated tumour cell dormancy [26]. For instance, TNF-α-induced NFκB-p65 signalling increases survival in both ER+/HER2- BC and TNBC cells in vitro [24]. NFκB also directly upregulates CXCR4 expression in BC cells, thereby increasing BC cell mobility and metastasis [56]. Furthermore, NFκB-mediated CXCR4 expression appears to promote BC metastasis, and BC cell colonisation of bone in particular [57]. In terms of uPAR, NFκB regulates both uPA and uPAR in ER+ BC and TNBC cells, which together promote invasiveness [58]. These results seminally suggest that NFκB-mediated PEDF downregulation under post-menopausal oestrogen conditions in BC may enhance activation of pro-metastatic pathways, which may be relevant to BC bone metastasis in post-menopausal BC patients.

When compared to treatment with a NFκB inhibitor alone, we found PEDF treatment alone demonstrated superior ability to reduce BC cell viability and downregulate pro-metastatic biomarkers under post-menopausal conditions in ER+/HER2- BC, and under pre-menopausal conditions in TNBC cells. This is surprising as we expected combining NFκB inhibition and PEDF treatment to have a synergistic effect on downregulating pro-metastatic biomarkers and metastatic function. Interestingly, JSH-23 (±PEDF) treatment increased MCF-7 viability under post-menopausal oestrogen conditions, which may be related to time-dependent effects of NFκB-mediated regulation of BC cell proliferation and apoptosis in the presence of oestrogen. A previous study indicates that while E2 initially blocks NFκB activation leading to enhanced BC cell proliferation, prolonged E2 exposure (48 h) has the opposite effect, increasing NFκB activation and subsequent BC cell apoptosis [59]. Furthermore, this same study found that JSH-23 treatment (20 µM) blocked NFκB-mediated apoptosis. As the cells in our studies were exposed to oestrogen-containing media for a similar length of time and treated with the same NFκB inhibitor at the same dose, it’s possible that JSH-23 (±PEDF) treatment blocked NFκB-mediated apoptosis, resulting in the observed increase in BC cell viability. Further time-course experiments coupled with expression analysis and functional studies (including apoptosis and proliferation assays), could further elucidate the time-dependent effects of JSH-23/PEDF treatment on NFκB expression under pre- and post-menopausal oestrogen conditions. These results may also signify PEDF downregulates NFκB, as well as TNFα, CXCR4, and uPAR expression to a greater extent than the NFκB functional inhibitor used, at least under the present experimental conditions, possibly via regulating additional NFκB-dependent and -independent pathways, leading to greater downstream effects on cell proliferation than NFκB inhibition. Alternatively, PEDF treatment may have had more direct effects on regulating other pathways critical to cell cycle progression via NFκB-independent pathways, as indicated by previous studies in fibroblasts [60]. In line with our initial hypotheses, combined treatment with PEDF and NFκB inhibitor reduced colony formation, and therefore anchorage-independent growth of ER+/HER2- BC cells under post-menopausal oestrogen conditions. This indicates that PEDF-mediated NFκB-p65 downregulation coupled with specific NFκB-65 inhibition has the greatest effect on downregulating NFκB-p65-mediated signalling central to colony formation. Given this synergistic effect was seen under post-menopausal oestrogen conditions specifically in ER+ BC cells, this may indicate that E1-mediated NFκB-p65 upregulation/overactivation is a key feature of ER+ BC metastasis post-menopause and this may be less central to TNBC metastasis. Altogether, the present data point towards the PEDF/NFκB reciprocal regulatory axis as a key factor underpinning PEDF anti-metastatic action in pre-menopausal TNBC and post-menopausal ER+/HER2- BC cells that has not been previously described.

Our previous studies highlighted PEDF downregulation in the presence of post-menopausal oestrogen levels in both ER+/HER2- BC and TNBC cells (manuscript submitted), contrary to expectations that higher circulating E2 levels pre-menopause [61] would have a greater downregulatory effect on PEDF, based on previous studies investigating E2 and PEDF in other tissues [14,62,63]. In line with our previous study, we suggest that in fact, higher E1/E2 intratumoral ratios present post-menopause [35] are associated with increased intracellular metastatic biomarker expression as opposed to higher E2/E1 intratumoral ratios present in pre-menopausal ER+ BC. In ER+/HER2- BC, PEDF had greater anti-metastatic action in the presence of circulating E1, E2, and E1S levels associated with post-menopause, likely because the corresponding intratumoral oestrogen ratios and associated signalling pathways downregulate PEDF to a greater extent than those present in pre-menopausal BC.

Given that E1 upregulates NFκB and E2 downregulates NFκB in ER+/HER2- BC cells [38], and that a reciprocal regulatory axis may exist between PEDF/NFκB, we suspect E1-driven p-NFκB-p65 upregulation/activation is responsible, at least in part, for the PEDF downregulation and associated enhanced metastatic potential observed in ER+/HER2- BC cells under post-menopausal oestrogen conditions. The post-menopausal environment in bone is associated with inflammation [64], immunosuppression [65], and bone loss [66], which altogether promote a tumour microenvironment (TME) and a pre-metastatic niche permissive for BC bone metastasis. We propose that the post-menopausal hormonal milieu and subsequent intratumoral E1 levels, are associated with a pro-inflammatory phenotype in ER+/HER2- BC, characterised by increased TNFα and NFκB expression, which work in tandem to downregulate PEDF expression, thus increasing BC metastatic potential as PEDF’s control over pro-metastatic mediators is lost. As such, replenishing this PEDF loss associated with post-menopausal oestrogen levels, may reduce ER+/HER2- BC metastasis and account for the results observed here. Our previous studies found PEDF is similarly downregulated under post-menopausal conditions in TNBC cells (manuscript submitted), therefore the rescuing effect described for ER+/HER2- BC may not explain the enhanced anti-metastatic action observed under pre-menopausal conditions in TNBC. Consequently, PEDF-mediated p-NFκB-p65 downregulation may also be important for PEDF anti-metastatic function in pre-menopausal TNBC. p-NFκB-p65 expression was higher in the presence of pre-menopausal oestrogens in TNBC cells (Appendix A). This may mean that PEDF treatment of TNBC cells is able to reduce p-NFκB-p65 and subsequent uPAR expression, thus accounting for improved anti-metastatic action of PEDF observed in TNBC cells under pre-menopausal oestrogen conditions. A summary of proposed mechanisms driving PEDF expression and function under pre- versus post-menopausal oestrogen conditions in ER+/HER2- BC and TNBC cells is provided in Figure 6.

Our study explored the effects of PEDF on pro-metastatic mediators and function in a clinically relevant in vitro system, which included the major circulating oestrogens at physiologically relevant concentrations, as reported in pre- and post-menopausal BC patients [35]. Previous studies investigating oestrogen and PEDF focused solely on E2, while E2 is an important bioactive oestrogen, such models do not represent the complexity of the oestrogen milieu present in pre- and post-menopausal BC patients. Our group is the first to develop an in vitro model system that reflects clinically relevant oestrogen levels reported in pre- and post-menopausal BC patients. E2 and E1 are the major circulating bioactive oestrogens pre- and post-menopause, respectively; with E1S the most abundant circulating oestrogen, and an important oestrogen precursor post-menopause [67,68]. Organic anion transport proteins (OATPs) facilitate E1S cellular uptake, while E1 and E2 bind intracellular ERs. Expression of these transport proteins, receptors, and enzymes that convert biologically inactive E1S to bioactive E1 and E2, can be overexpressed in ER+ BC [68,69,70,71] and in TNBC [72,73,74], where oestrogen signalling occurs via ‘imperfect oestrogen response elements’ [75]. Plasma oestrogens specifically, and plasma E1 in particular, appear to be central to determining intratumoral oestrogen levels [35,61]. In addition to including physiologically relevant circulating oestrogens, collagen-coated plates were used to simulate the TME in vitro. Collagen is an essential component of bone and TMEs, which plays a crucial role in metastasis [23], and also mediates PEDF cellular effects [76,77]. As such, growing well-characterised BC cell lines on collagen, with major circulating oestrogens at levels pertinent to menopausal status, allows investigation of PEDF in a menopausal in vitro model, clinically and biologically relevant to the bone TME in BC patients. As changing circulating oestrogen levels are associated with menopause and a link between E2 and PEDF has already been established [14,63,78], our study focused on circulating E2, as well as E1 and E1S levels. As other sources of oestrogen and oestrogen precursors also impact intratumoral oestrogen levels, future studies should address the effect of other circulating oestrogen precursors, as well as local oestrogen synthesis on intratumoral oestrogens and PEDF anti-metastatic effects.

The TME is a complex and dynamic space, notoriously difficult to replicate outside a living organism. The TME is typically characterised by inflammation and oxidative stress, with interactions between extracellular matrix, stromal cells, immune cells, and tumour cells driving tumorigenesis and metastasis. Therefore, progressing this work to animal models of BC metastasis, that potentially also model menopause-associated bone loss, could provide a deeper understanding of the clinical significance of these findings in post-menopausal BC bone metastasis. We explored PEDF in ER+/HER2- and TNBC subtypes in this study, however this work could be extended to investigate PEDF in HER2+ BC, as menopausal status may affect HER2 status in BC [79]. Furthermore, this work would also benefit from being replicated in other BC cell lines, as well as quantification of PEDF and pro-metastatic mediators at the transcriptional level.

**Figure 6 ijms-23-15641-f006:**
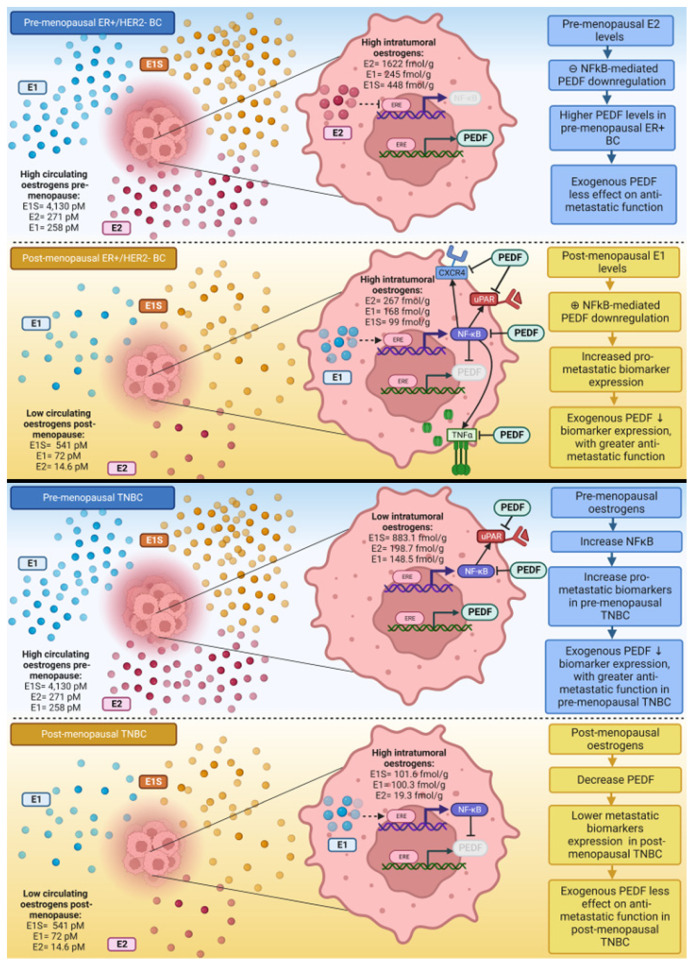
**Summary of proposed mechanisms underlying differential PEDF expression and function under pre- versus post-menopausal circulating oestrogen conditions in ER+/HER2- and TNBC cells.** Circulating and intratumoral oestrogen levels in pre- and post-menopausal oestrogen receptor (ER)+ and ER- breast cancer patients as reported by Lønning et al. [35].

## 4. Materials and Methods

### 4.1. Reagents

0.5% Trypsin-EDTA 10X [Gibco, Waltham, MA, USA, Cat # 15400-054], antibiotic antimycotic solution 100X stabilised [Sigma, St. Louis, MO, USA, Cat # A5955], Bovine serum albumin (Sigma Cat # A2153), CellTitre Blue Reagent [Cat # G808B], Collagen [Sigma, Cat # C8919], Dulbecco’s Modified Eagle Medium 1x [Gibco, Cat # 24063-029], DPX with colourfast mounting solution [Cat # 11023DPX], Dulbecco’s phosphate-buffered saline [BioWhittaker, Waltham, MA, USA, Cat # 17-512F], E1 [Cat # 10006485, Cayman chemicals], E1S [Cat # 20513 Cayman chemicals], E2 (β-Oestradiol) [Sigma, Cat # E2758], Fetal bovine serum (FBS) [Bovogen Biologicals, Keilor East, Australia, Cat # SFBS-F], glycerol [Sigma-Aldrich Cat # G6279], goat anti-mouse secondary antibody biotin [Invitrogen, Waltham, MA, USA, 62-6540], goat serum (normal) [DAKO, Carpinteria, CA, USA, Cat # X0907], Opti-MEM reduced serum medium, phenol red free media [Gibco, Cat # 11058021], polyclonal goat anti-rabbit immunoglobulins/biotinylated [DAKO, Cat # E0432], primary antibodies(CXCR4 12G5 [Santa Cruz, Santa Cruz, CA, USA, SC-12764]; MT1-MMP [Santa Cruz, SC-26703]; p-NFκB-p65 27.Ser 536 [Santa Cruz, SC-136548]; RANKL G-1 [Santa Cruz, SC-377079]; TNF-α [Santa Cruz, SC-1348]; uPA [Santa Cruz, SC-14019]; uPAR [Santa Cruz, SC-10815]), rabbit anti-goat IgG biotin conjugate [Novex, Waltham, MA, USA, A16146], recombinant PEDF [Bioproducts MD, Oakdale, MN, USA, Cat # PEDF-500], saponin [Sigma, Cat # 47036-50G-F], SigmaFast DAB with metal enhancer [Sigma, Cat # D0426], a [Cat # PK-6100], α-PEDF antibody [Bioproducts MD, Cat # AB-PEDF 1].

### 4.2. Ethics

As this was a negligible risk research study, Hollywood Private Hospital Human Research Ethics Committee (HREC) had previously approved that no study-specific consent was necessary to access breast tissue of deceased patients. Patients had previously provided written consent for use of pathological specimens (collected as part of their standard of care) stored at Helen Sewell Tumour Bank (HSTB). Patient informed consent was obtained from all living patients to access bone tissue. All patients had provided consent for use of their de-identified medical information for research purposes. This project was approved by Hollywood Private Hospital (reference HPH519) and Curtin University (reference HRE2018-0189) HRECs.

### 4.3. Patient Tissue Samples

Samples from patients who received BC treatment at Mount Hospital and Hollywood Private Hospital under the care of co-author (AC) between 2005 and 2020 were used in this study. Tissue was obtained from the HSTB via the Breast Cancer Research Centre-WA. Breast slides were obtained from untreated breast tissue and bone metastatic specimens were collected prior to commencing metastatic treatment. All samples were archival primary BC and/or metastases specimens as formalin-fixed paraffin-embedded (FFPE) blocks. All bone samples were de-calcified prio to FFPE processing using standardised and validated techniques by a commercial diagnostic lab (Western Diagnostic Pathology) as described previously [19]. All sections were reviewed by study pathologist (JG) to confirm sections contained representative tumour tissue. All sections (4 µm) were prepared on a Leica Biosystems RM2245 Semi-Automated Rotary Microtome. Normal stomach and renal tissue obtained from healthy controls were used as internal positive controls. PEDF status was known for all specimens as per our previous study [19].

### 4.4. Immunohistochemistry

Immunohistochemistry (IHC) staining was initiated by deparaffinising sections using xylene and rehydrating in decreasing ethanol concentrations (100%, 95%, 70%, 30%). Slides were washed in PBS with 0.1% Triton X-100 prior to performing antigen retrieval by microwaving slides in high pH buffer (10 mM Tris/1 mM EDTA/PBS) for 12 min at high heat (95 °C). Slides were allowed to cool to room temperature (RT) before blocking endogenous peroxidase activity by incubating slides in 0.3% hydrogen peroxide/dH_2_O for 30 min. Blocking in 10% normal serum/0.25% BSA for 30 min was then performed. Slides were then incubated with primary antibody (p-NFκB-p65, 1:100) for 3 h RT. Slides were then washed and incubated with biotinylated goat anti-rabbit secondary antibody (1:1000) for 30 min. Slides were rinsed with PBS and avidin/biotinylated HRP prepared and applied as per manufacturer’s instructions. DAB solution was prepared in the dark immediately before use and applied to slides for 3–5 min until a blue colour change was detected. Slides were then rinsed and dehydrated by incubating in 70% ethanol, then 100% ethanol. Slides were cleared with 100% xylene and mounted with mounting solution and sealed with coverslip. Images were collected under light microscopy at 200X magnification.

### 4.5. Scoring of IHC

Score analysis was performed as previously described [19]. Briefly, staining in subcellular locations was scored by two blinded observers by combining staining intensity and distribution [18]. Our scoring system combined a staining intensity score (1 = weak, 2 = moderate, 3 = strong) and a percentage positive cells score (0% = 0; 1–25% = 1; 26–50% = 2; 51–75% = 3; 76–100% = 4), to give an overall score to determine PEDF/NFκB-p65 expression status (0–5 = Negative, 6–7 = Positive), thus facilitating semi-quantitative analysis following methods described previously [18,36,78].

### 4.6. Cell Culture

MDA-MB-231 and MCF-7 cells were obtained from ATCC and maintained in standard tissue culture throughout all experiments. All assays were performed on collagen-coated (3 µg collagen/well) well plates, as described previously [80]. For experiments, exponentially growing cells were trypsinised and cell pellets re-suspended in low serum Opti-MEM media with 1% FBS, 1% antibiotic/antimycotic and either pre- or post-menopausal oestrogens. Briefly, menopausal conditions were replicated in vitro by supplementing media with circulating concentrations of oestrogens as reported in pre-menopausal (E1: 258 pM; E1S: 4,130 pM; E2: 271 pM) and post-menopausal (E1: 72 pM; E1S: 541 pM; E2: 14.6 pM) BC patients [35]. All experiments were performed in quadruplicate, with multiple passages of each cell line where possible, and repeated. Cells were treated with either recombinant PEDF (100 nM, physiologically relevant level of PEDF in normal circulation [81]), and/or NFκB inhibitor JSH-23 (20 µM) both prepared in serum-free Opti-MEM, or serum-free Opti-MEM as a negative control. JSH-23 has been previously shown to block NFκB expression and activity in MCF-7 cells at 20 µM [59]. Similarly, JSH-23 has been shown to inhibit NFκB expression in MDA-MB-231 cells [82,83].

### 4.7. Immunocytochemistry

ICC experiments and quantitative analysis of staining intensity was performed as previously described (manuscript submitted). Briefly, cells were seeded (MCF-7 at 10,000 cells/mL, MDA-MB-231 at 5000 cells/mL) on collagen-coated 96-well plates in media supplemented with pre- or post-menopausal oestrogens and incubated overnight. Cells were treated with PEDF, and/or JSH-23, or control for 24-h. Following fixation, permeabilisation, and blocking, cells were incubated with primary antibodies (all biomarkers 1:250; PEDF 1:500) overnight at 4 °C. NFκB-p65 phosphorylated at serine 536 [p-NFκB-p65 (SER 536)] is important for NFκB activation, nuclear translocation, and transcription of target genes [84], and interacts with PEDF [85]. Furthermore, p-NFκB-p65 (SER 536) overexpression in primary BC is associated with poorer patient survival, higher tumour grade, and disease recurrence [24]. Negative control wells were incubated with PBS only, and all staining was normalised to negative controls.

### 4.8. Statistical Analysis

Unpaired student t-test (to compare the difference between 2 means) and one-way ANOVA (to compare the difference between >3 means) was used to compare means in vitro data sets via Graphpad Prism version 8 software. Briefly, the mean and standard deviation were calculated and compared for control and treated groups in pre- or post-menopausal conditions. For IHC stained tissue samples, unpaired student t-test was used to compare the mean overall staining scores.

## 5. Conclusions

These results provide new mechanistic insight regarding the role of pre- and post-menopausal levels of oestrogens in regulating PEDF expression and function in metastatic BC. We report for the first time a potential reciprocal regulatory relationship between PEDF and p-NFκB-p65, which may be a key mechanism driving differential PEDF expression and anti-metastatic function in pre-menopausal TNBC and post-menopausal ER+/HER2- BC. Metastatic BC remains incurable and improved bone-targeting therapies are desperately sought. Given the well-established anti-tumour function of PEDF in breast and bone, PEDF represents a therapeutic candidate. Our findings show for the first time that the mechanisms underlying PEDF anti-metastatic function in BC are differentially governed by not only E2, but also E1 and E1S at levels circulating in pre- and post-menopausal BC patients in a molecular subtype-specific manner. Altogether these results support the premise that PEDF treatment reduces pro-metastatic phenotypes of TNBC cells under pre-menopausal oestrogen conditions and ER+/HER2- BC cells under post-menopausal oestrogen conditions, which may be relevant for the development of BC bone metastases. Our data indicate BC molecular subtype and menopausal status are relevant to future clinical PEDF research. We further highlight pre-menopausal TNBC patients and post-menopausal ER+/HER2- BC patients as target populations for future therapeutic PEDF development.

## Figures and Tables

**Figure 1 ijms-23-15641-f001:**
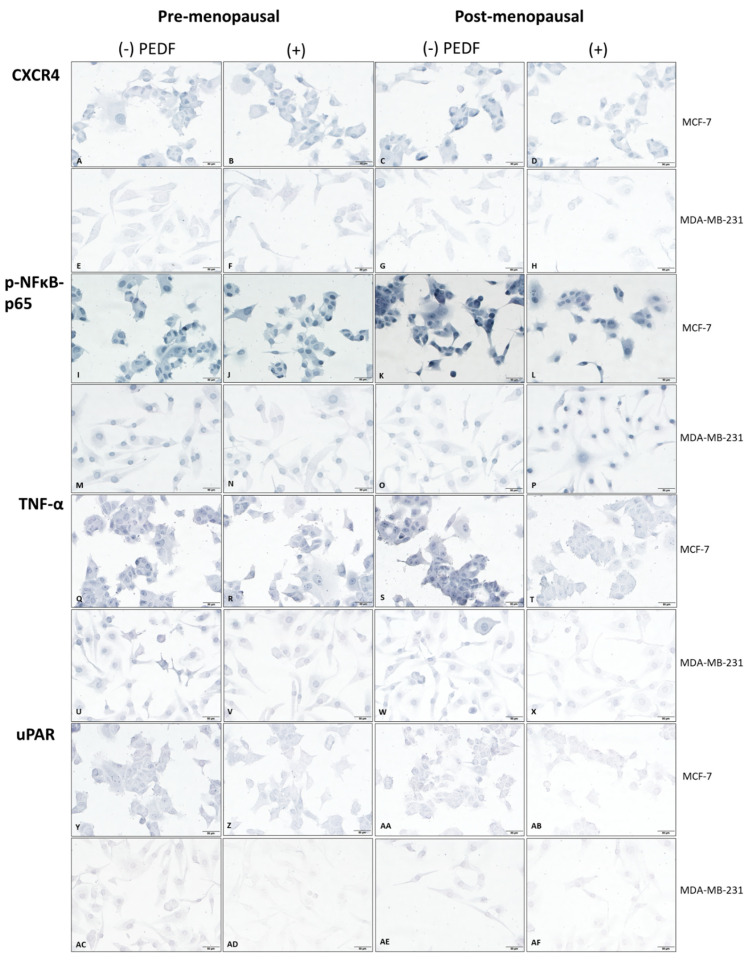
**Biomarker immunocytochemistry in MCF-7 and MDA-MB-231 cells following PEDF treatment in pre- versus post-menopausal oestrogens**. Biomarker expression via immunocytochemistry staining (blue) in MCF-7 and MDA-MB-231 cells following incubation in media supplemented with either pre-menopausal (E1: 258 pM; E1S: 4130 pM; E2: 271 pM) or post-menopausal (E1: 72 pM; E1S: 541 pM; E2: 14.6 pM) oestrogen levels and treated with recombinant PEDF (100 nM) or control for 24-h. CXCR4 staining under pre-menopausal conditions in MCF-7 treated with control (**A**) or PEDF (**B**) and in MDA-MB-231 treated with control (**E**) or PEDF (**F**), or under post-menopausal conditions in MCF-7 treated with control (**C**) or PEDF (**D**) and in MDA-MB-231 treated with control (**G**) or PEDF (**H**). Phosphorylated (p-)NFκB-p65 staining under pre-menopausal conditions in MCF-7 treated with control (**I**) or PEDF (**J**) and in MDA-MB-231 treated with control (**M**) or PEDF (**N**), or under post-menopausal conditions in MCF-7 treated with control (**K**) or PEDF (**L**) and in MDA-MB-231 treated with control (**O**) or PEDF (**P**). TNF-α staining under pre-menopausal conditions in MCF-7 treated with control (**Q**) or PEDF (**R**) and in MDA-MB-231 treated with control (**U**) or PEDF (**V**), or under post-menopausal conditions in MCF-7 treated with control (**S**) or PEDF (**T**) and in MDA-MB-231 treated with control (**W**) or PEDF (**X**). uPAR staining under pre-menopausal conditions in MCF-7 treated with control (**Y**) or PEDF (**Z**) and in MDA-MB-231 treated with control (**AC**) or PEDF (**AD**), or under post-menopausal conditions in MCF-7 treated with control (**AA**) or PEDF (**AB**) and in MDA-MB-231 treated with control (**AE**) or PEDF (**AF**). Biomarker staining intensity in cells treated with control or PEDF for CXCR4 (**a**), p-NFκB-p65 (**c**), TNF-α (**e**), and uPAR (**g**). Biomarker fold change in PEDF-treated cells compared to control in pre- or post-menopausal conditions for CXCR4 (**b**), p-NFκB-p65 (**d**), TNF-α (**f**), and uPAR (**h**). Scale bar 50 µM. **** *p* < 0.0001; *** *p* = 0.0001–0.0009; ** *p* = 0.001–0.009; * *p* = 0.01–0.05.

**Figure 2 ijms-23-15641-f002:**
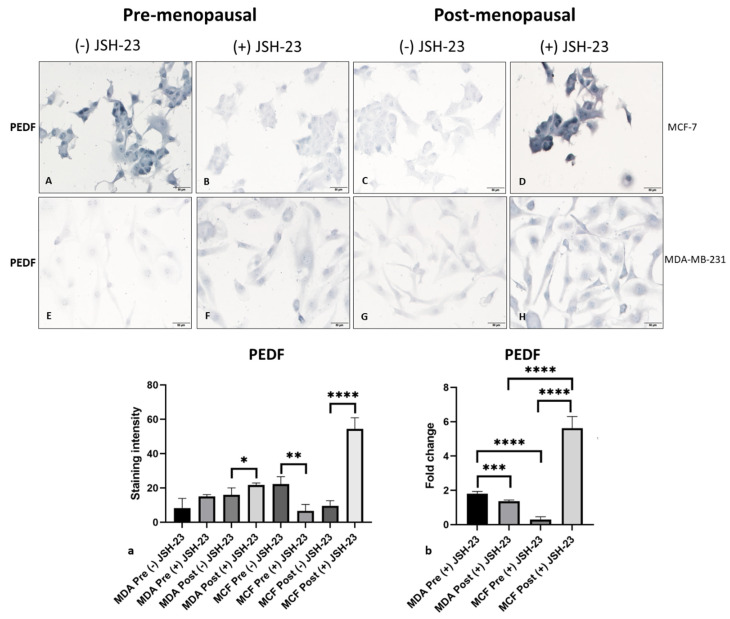
**PEDF expression in MCF-7 and MDA-MB-231 cells following JSH-23 treatment in pre- versus post-menopausal oestrogens.** PEDF expression determined via immunocytochemistry staining (blue) in cells following incubation in media supplemented with either pre-menopausal (E1: 258 pM; E1S: 4130 pM; E2: 271 pM) or post-menopausal (E1: 72 pM; E1S: 541 pM; E2: 14.6 pM) levels of oestrogens and treated with NFκB-p65 inhibitor JSH-23 (20 µM) or control for 24-h. PEDF staining under pre-menopausal conditions in MCF-7 treated with control (**A**) or JSH-23 (**B**) and in MDA-MB-231 treated with control (**E**) or JSH-23 (**F**), and under post-menopausal conditions in MCF-7 treated with control (**C**) or JSH-23 (**D**) and in MDA-MB-231 treated with control (**G**) or JSH-23 (**H**). PEDF staining intensity (total intensity (gray) of cells/total pixels of cells) in cells treated with control versus JSH-23 (**a**). PEDF expression fold change in JSH-23-treated cells compared to control in pre- or post-menopausal conditions for (**b**). Scale bar 50 µM. **** *p* < 0.0001; *** *p* = 0.0001–0.0009; ** *p* = 0.001-–0.009; * *p* = 0.01-–-0.05.

**Figure 3 ijms-23-15641-f003:**
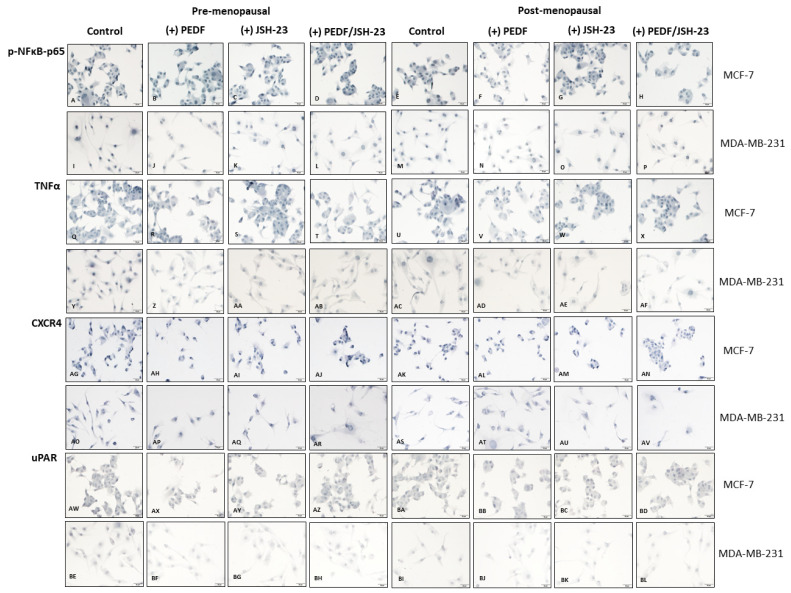
**Biomarker expression following PEDF and JSH-23 treatment in MCF-7 and MDA-MB-231 cells in pre- and post-menopausal conditions.** Biomarker expression via immunocytochemistry staining (blue) in cells following incubation in pre- (E1: 258 pM; E1S: 4130 pM; E2: 271 pM) or post-menopausal (E1: 72 pM; E1S: 541 pM; E2: 14.6 pM) levels of oestrogens and treatment with PEDF (100 nM), JSH-23 (20 µM), PEDF (100 nM) + JSH-23 (20 µM), or control for 24-h. Phosphorylated (p-)NFκB-p65 expression under pre-menopausal conditions in MCF-7 treated with control (**A**), PEDF (**B**), JSH-23 (**C**), or PEDF + JSH-23 (**D**) and in MDA-MB-231 treated with control (**I**), PEDF (**J**), JSH-23 (**K**), or PEDF + JSH-23 (**L**), and under post-menopausal conditions in MCF-7 treated with control (**E**), PEDF (**F**), JSH-23 (**G**), or PEDF + JSH-23 (**H**) and in MDA-MB-231 treated with control (**M**), PEDF (**N**), JSH-23 (**O**), or PEDF + JSH-23 (**P**). TNF-α expression under pre-menopausal conditions in MCF-7 treated with control (**Q**), PEDF (**R**), JSH-23 (**S**), or PEDF + JSH-23 (**T**) and in MDA-MB-231 treated with control (**Y**), PEDF (**Z**), JSH-23 (**AA**), or PEDF + JSH-23 (**AB**), or under post-menopausal conditions in MCF-7 treated with control (**U**), PEDF (**V**), JSH-23 (**W**), or PEDF + JSH-23 (**X**) and in MDA-MB-231 treated with control (**AC**), PEDF (**AD**), JSH-23 (**AE**), or PEDF + JSH-23 (**AF**). CXCR4 expression under pre-menopausal conditions in MCF-7 treated with control (**AG**), PEDF (**AH**), JSH-23 (**AI**), or PEDF + JSH-23 (**AJ**) and in MDA-MB-231 treated with control (**AO**), PEDF (**AP**), JSH-23 (**AQ**), or PEDF + JSH-23 (**AR**) or under post-menopausal conditions in MCF- treated with control (**AK**), PEDF (**AL**), JSH-23 (**AM**), or PEDF + JSH-23 (**AN**) and in MDA-MB-231 treated with control (**AS**), PEDF (**AT**), JSH-23 (**AU**), or PEDF + JSH-23 (**AV**). uPAR expression under pre-menopausal conditions in MCF-7 treated with control (**AW**), PEDF (**AX**), JSH-23 (**AY**), or PEDF + JSH-23 (**AZ**) and in MDA-MB-231 treated with control (**BE**), PEDF (**BF**), JSH-23 (**BG**), or PEDF + JSH-23 (**BH**) or under post-menopausal conditions in MCF-7 treated with control (**BA**), PEDF (**BB**), JSH-23 (**BC**), or PEDF + JSH-23 (**BD**) and in MDA-MB-231 treated with control (**BI**), PEDF (**BJ**), JSH-23 (**BK**), or PEDF + JSH-23 (**BL**). Biomarker staining intensity in cells treated with control, PEDF, JSH-23, or PEDF + JSH-23 for p-NFκB-p65 (**a**) TNF-α (**c**), uPAR (**e**), and CXCR4 (**g**). Biomarker fold change in PEDF-treated cells versus control in pre- or post-menopausal conditions for p-NFκB-p65 (**b**), TNF-α (**d**), uPAR (**f**), and CXCR4 (**h**). Scale bar 50 µM.

**Figure 4 ijms-23-15641-f004:**
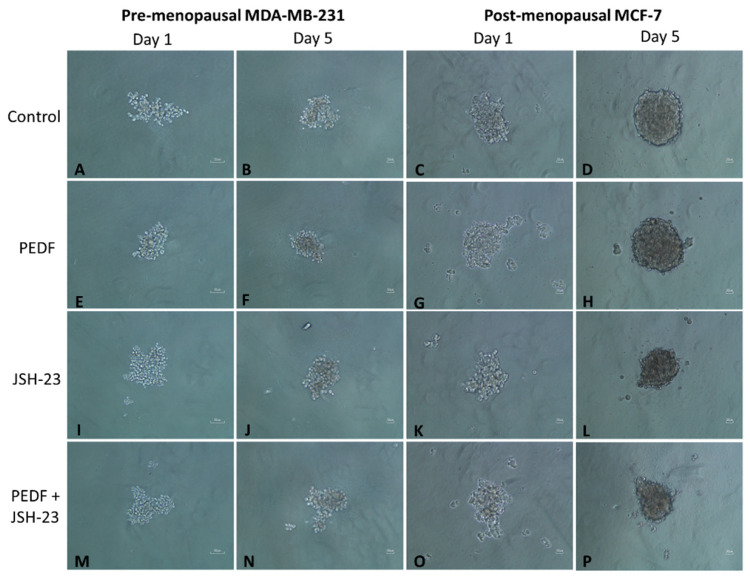
**Functional assays in MCF-7 and MDA-MB-231 following PEDF and/or JSH-23 treatment in pre- versus post-menopausal conditions.** Cells were incubated in media supplemented with pre- (E1: 258 pM; E1S: 4130 pM; E2: 271 pM) or post-menopausal (E1: 72 pM; E1S: 541 pM; E2: 14.6 pM) oestrogen levels and treated with PEDF (100 nM), JSH-23 (20 µM), combined PEDF (100 nM) + JSH-23 (20 µM), or control for 5 days for colony formation assays and for 24 h for viability assays. Fold change in colony size compared to day 1 replicates was calculated for MDA-MB-231 under pre-menopausal conditions and MCF-7 under post-menopausal conditions. Microscope images of MDA-MB-231 colonies at day 1 under pre-menopausal conditions treated with control (**A**), PEDF (**E**), JSH-23 (**I**), or PEDF + JSH-23 (**M**) and at day 5 treated with control (**B**), PEDF (**F**), JSH-23 (**J**), or PEDF + JSH-23 (**N**), and MCF-7 colonies under post-menopausal conditions at day 1, treated with control (**C**), PEDF (**G**), JSH-23 (**K**), or PEDF + JSH-23 (**O**) and at day 5 treated with control (**D**), PEDF (**H**), JSH-23 (**L**), or PEDF + JSH-23 (**P**). Graphs showing fold change colony growth following 5 days treatment for MDA-MB-231 cells under pre-menopausal conditions (**Q**) and MCF-7 cells under post-menopausal conditions (**R**). CellTitre Blue assay was used to determine viability of MDA-MB-231 under pre-menopausal conditions (**S**) and MCF-7 under post-menopausal conditions (**T**). Scale bar 50 µM. ** *p* = 0.001–0.009; * *p* = 0.01–0.05.

**Figure 5 ijms-23-15641-f005:**
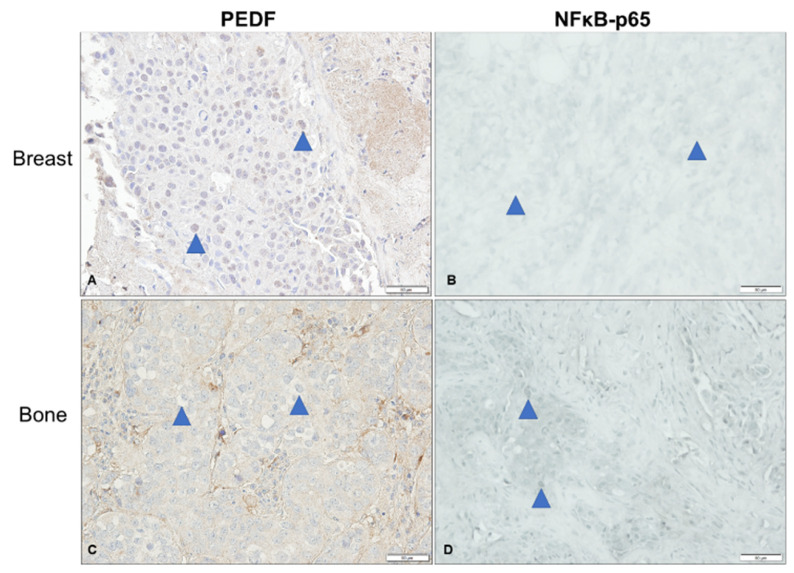
**Immunohistochemistry analysis of PEDF and p-NFκB-p65 expression in ER+/HER2- breast cancer and bone metastases tissue.** Microscope images of untreated oestrogen receptor positive (ER+)/human epidermal growth factor receptor-2 negative (HER2-) primary breast cancer (BC) and bone metastases patient tissue specimens, stained via immunohistochemistry (IHC) for phosphorylated (p-)NFκB-p65 expression (blue stain), compared with PEDF expression (brown stain) of a small subset of samples as previously determined by our group [19]. PEDF expression was determined in the same cohort of tissue specimens as previously published by our group [19], here we revisited a small sub-set of these samples and additionally stained for NFκB-p65 via IHC using the same scoring technique previously described [19]. PEDF (brown) stained BC cells in breast (**A**) and bone (**C**), and NFκB-p65 stained (blue) in BC cells in breast (**B**) and bone (**D**). Blue arrows indicate invasive BC cells. Scale bar, 50 µM.

**Table 1 ijms-23-15641-t001:** Summary of biomarker function in breast cancer bone metastasis.

Biomarker	Function in Breast Cancer Bone Metastasis	References
CXCR4	Promotes BC cell homing to bone; enhances BC cell proliferation and survival; increases neovascularisation; drives chemoresistance.	[20,21,22]
MT1-MMP	Promotes ECM proteolysis and BC cell invasion	[23]
NFκB	Promotes osteoclastogenesis; promotes BC cell survival; promotes DTC dormancy; promotes BC cell invasion; drives EMT.	[24,25,26,27]
RANKL	Promotes osteoclastogenesis; promotes osteolysis; Promotes BC cell homing to bone;	[28,29]
TNFα	Promotes DTC colonisation and survival in bone; promotes osteoclastogenesis and osteolysis.	[30,31]
uPA/uPAR	Promotes ECM proteolysis and BC cell invasion; promotes DTC dormancy	[32,33,34]

*BC: breast cancer; CXCR4: C-X-C motif chemokine receptor 4; DTC: disseminated tumour cell; ECM: extracellular matrix; EMT: epithelial-mesenchymal transition; MT1-MMP: membrane-type 1 matrix metalloproteinase; NFκB: nuclear factor kappa B; RANKL: receptor activator of nuclear factor kappa-Β ligand; TNFα: tumour necrosis factor; uPA: urokinase-type plasminogen activator; uPAR: urokinase-type plasminogen activator receptor.*

**Table 2 ijms-23-15641-t002:** PEDF and p-NFκB scores (mean ± standard deviation) of ER+/HER2- breast cancer and bone metastases specimens.

Tumour Site and Subcellular Localisation	Overall Scores (Mean ± Standard Deviation)		*p*-Value
Breast (*n* = 4)	PEDF	p-NFκB-p65	
− Cytoplasm	3.25 ± 0.96	3 ± 2.45	0.8555
− Nuclear	2.75 ± 1.5	2.5 ± 1.91	0.8439
**Bone (*n* = 4)**			
− Cytoplasm	3 ± 0.82	2.5 ± 3	0.7586
− Nuclear	1.5 ± 1	4.75 ± 2.22	**0.0369 ***

*PEDF: pigment epithelium-derived factor; p-NFκB-p65: phosphorylated nuclear factor κappa B-p65 subunit. Mean difference was assessed using unpaired-sample t-test; * Statistically significant at alpha = 0.05.*

## Data Availability

The datasets generated during and/or analysed during the current study are available from the corresponding author on reasonable request.

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
