# Peer review of "NFκB-Mediated Mechanisms Drive PEDF Expression and Function in Pre- and Post-Menopausal Oestrogen Levels in Breast Cancer"

_ijms, 2022, doi:10.3390/ijms232415641_

Round 1

Reviewer 1 Report

The authors have demonstrated how PEDF could be a potential target for anti-tumor therapy for both bone and breast cancers. The author showed how pre and post-menopausal oestrogen levels regulate PEDF expression. PEDF treatment reduces levels of NFkB, TNF, CXCR4, and other factors in both cancer cells. PEDF treatment downregulates the expression of tumor biomarkers in post-menopausal conditions in MCF7 and pre-menopausal conditions in MDA MB 231 cells. I have a few suggestions as mentioned below:

1. In the results, the Author has mentioned that PEDF treatment downregulates NFKB expression under pre-menopausal conditions in TNBC cells. Whereas, CXCR4 is upregulated in TNBC cells under the same conditions. Why CXCR4 is upregulated under PEDF treatment/ CXCR4 is a metastatic gene and PEDF has an anti-metastasis effect.

2. In Figure 1, remove space in between NFKB and all figure legends should be together with the figure titles.

3. Author can explain in the results why they have chosen CXCR4, TNF, and other factors for metastasis studies. 

4.  An NFKB inhibitor- JSH-23 is used to study the PEDF and NFKB regulation mechanism. The author should show the NFKB expression under JSH-23 treatment.

5. In MCF7, colony growth is reduced with JSH23 and PEDF treatment whereas cell viability is induced. Why?

6. Figure 6 is missing in the manuscript.

Author Response

General comment: The authors have demonstrated how PEDF could be a potential target for anti-tumor therapy for both bone and breast cancers. The author showed how pre and post-menopausal oestrogen levels regulate PEDF expression. PEDF treatment reduces levels of NFkB, TNF, CXCR4, and other factors in both cancer cells. PEDF treatment downregulates the expression of tumor biomarkers in post-menopausal conditions in MCF7 and pre-menopausal conditions in MDA MB 231 cells. I have a few suggestions as mentioned below:

  • Reviewer comment: In the results, the Author has mentioned that PEDF treatment downregulates NFKB expression under pre-menopausal conditions in TNBC cells. Whereas, CXCR4 is upregulated in TNBC cells under the same conditions. Why CXCR4 is upregulated under PEDF treatment/ CXCR4 is a metastatic gene and PEDF has an anti-metastasis effect.

Author response: The functional/physiological significance of the slight increase in CXCR4 expression observed following PEDF treatment in TNBC (and MCF-7) cells under pre-menopausal conditions is unknown. However as this did not reach statistical significance in any of our experimental repeats (Fig 1a), we may assume that the physiological significance of this is low. Please note that for this reason, this slight increase in CXCR4 expression was not initially discussed further in the manuscript. The reason for this slight increase in CXCR4 expression under pre-menopausal conditions in both cell lines is unknown, although this may be linked with the reduced anti-metastatic function of PEDF in MCF-7 under these conditions. The above additional comments have now been added to the manuscript.

  • Reviewer comment: In Figure 1, remove space in between NFKB and all figure legends should be together with the figure titles.

Author response: Thank you, spaces between figure titles and figure legends removed for all figures.

  • Reviewer comment: Author can explain in the results why they have chosen CXCR4, TNF, and other factors for metastasis studies.

Author response: Thank you for raising this, this information has been provided in context of the results observed in the discussion section. We have also added additional introductory information regarding each pro-metastasis biomarker summarised in table form in the results section, please see tracked changes on page 7-8.

  • Reviewer comment: An NFKB inhibitor- JSH-23 is used to study the PEDF and NFKB regulation mechanism. The author should show the NFKB expression under JSH-23 treatment.

Author response: Thank you for your suggestion, the inhibitor used has been previously well validated for inhibiting NFκB expression in MCF-7 cells, may I please draw your attention to page 6 of the materials and methods section where this is described. We have included additional information regarding the validation of this inhibitor in inhibiting NFκB expression in MDA-MB-231 cells also, please tracked changes on page 6.

  • Reviewer comment: In MCF7, colony growth is reduced with JSH23 and PEDF treatment whereas cell viability is induced. Why?

Author response: Thank you for this question, this result was certainly intriguing as we expected combined treatment with PEDF and JSH-23 to have a synergistic effect in reducing BC cell viability (as we saw with colony growth). We suspect these unexpected effects may be related to the time-dependent effects of NFκB-mediated regulation of BC cell proliferation/apoptosis in the presence of oestrogen. A previous study indicates that while E2 initially blocks NFκB activation leading to enhanced BC cell proliferation, prolonged E2 exposure (48h) has the opposite effect, increasing NFκB activation and subsequent BC cell apoptosis [1]. Furthermore, this same study found that blocking NFκB via JSH-23 at the same doses used in our study, blocked NFκB-mediated apoptosis. As the cells in our studies were exposed to oestrogen-containing media for the same length of time (seeded for 24h then treated for 24h, total exposure time to oestrogen-containing media = 48h) and treated with the same NFκB inhibitor at the same dose, it’s possible that JSH-23 (± PEDF) treatment blocked NFκB-mediated apoptosis, resulting in the observed increase in viability. The fact that PEDF can reduce BC cell viability whereas direct NFκB inhibition (via blocking NFκB nuclear translocation with JSH-23) increases viability may indicate that PEDF regulates BC cell viability via a different mechanism of NFκB inhibition or via alternative pathways. Future studies could further investigate this by determining the time-dependent effects of treatment on NFκB expression, by performing time-course experiments along with expression analysis and functional studies, including apoptosis and proliferation assays.

The comments above have been added to the discussion section of the manuscript, please see tracked changes on page 30

  • Reviewer comment: Figure 6 is missing in the manuscript.

Author response: Thank you, Figure 6 is provided to summarise our key findings and can be found after the discussion and before the conclusion, please see page 34 of the manuscript.

Reference

  1. Fan P, Tyagi AK, Agboke FA, Mathur R, Pokharel N, Jordan VC (2018) Modulation of nuclear factor-kappa B activation by the endoplasmic reticulum stress sensor PERK to mediate estrogen-induced apoptosis in breast cancer cells. Cell Death Discovery 4 (1):1-14  

Reviewer 2 Report

The manuscript entitled “NFκB-mediated mechanisms drive PEDF expression and function in pre-and post-menopausal oestrogen levels in breast cancer by Brook et al talks about the  pre-and post-menopausal bone microenvironments and the inverse relationship of PEDF and NF-kB. The results are good but the conclusions drawn from the data given are not fully assuring.

PEDF and NF-kB information is already known in the literature. I would request the authors to highlight the novality of the work before it can be accepted by the IJMS journal.

Specific comments below:

1)     Figure 1 does not give a clear picture of the data. The resolution of the images is low. The bar size given in the image is also not clear. Please enhance the image resolution.

2)     The images do not give a clear picture of the conclusions drawn by the authors. I would highly appreciate it if the authors perform a western blot to support their conclusions for at least the main proteins of the manuscript. The antibodies used for IHC  can be used for western blot too.

3)     Figure 2 is also the same. It is hard to conclude anything from it. The bars also do not tell the size.

4)     Please provide the bars in figure 4. Give the alphabet numbers to each graph. No statistical analysis was done for the MDA-MB-231 colony graph. Any reasons?

5)     Figure5: No bars

6)     Write tumor instead of tumour throughout the manuscript.

Author Response

General comment: The manuscript entitled “NFκB-mediated mechanisms drive PEDF expression and function in pre-and post-menopausal oestrogen levels in breast cancer by Brook et al talks about the  pre-and post-menopausal bone microenvironments and the inverse relationship of PEDF and NF-kB. The results are good but the conclusions drawn from the data given are not fully assuring.

PEDF and NF-kB information is already known in the literature. I would request the authors to highlight the novality of the work before it can be accepted by the IJMS journal.

Author response: Thank you for your suggestion, we have updated the introduction, discussion, and conclusion to emphasise the novelty of this work.

Specific comments below:

  • Reviewer comment: Figure 1 does not give a clear picture of the data. The resolution of the images is low. The bar size given in the image is also not clear. Please enhance the image resolution.

Author response: Thank you for your comments, the images within figure 1 have been enlarged as much as possible to increase the size of the scale bars/resolution and still allow the figure to fit within the margins of the page.

  • Reviewer comment: The images do not give a clear picture of the conclusions drawn by the authors. I would highly appreciate it if the authors perform a western blot to support their conclusions for at least the main proteins of the manuscript. The antibodies used for IHC can be used for western blot too.

Author response: We deliberately chose to use ICC as it allows us to visualise any intracellular compartmental changes to PEDF and other markers, which IB does not allow. As noted, NFkB is strongly stained in the nucleus, and some in the cytoplasm, a fact that IB would not be able to show unfortunately. Both IB and ICC are semi-quantitative, and we chose ICC due to its faster processing time and lack of stripping of membranes which can be problematic unless tightly controlled or repeated numerous times. These experiments have all been performed at least twice, with quadruplicate datasets, so we are confident of the trends (or lack thereof) demonstrated in our paper. In addition, the first author has since moved on to a different position, so running IBs is no longer possible, but as abovementioned, we chose ICC in the first place to see spatial differences on levels of PEDF and the selected biomarkers.

  • Reviewer comment: Figure 2 is also the same. It is hard to conclude anything from it. The bars also do not tell the size.

Author response:  Thank you for your comments, the images within figure 2 have been enlarged as much as possible to increase the size of the scale bars/resolution and still fit within the margins of the page.

  • Reviewer comment: Please provide the bars in figure 4. Give the alphabet numbers to each graph. No statistical analysis was done for the MDA-MB-231 colony graph. Any reasons?

Author response: The images in Figure 4 have been enlarged as much as possible to increase the size of the scale bars/resolution and still fit within the margins of the page. The alphabet labels are provided in the bottom left-hand corner of the images in figure 4 and are described in the figure legend underneath the figure. Statistical analysis was performed for all treated groups, no statistically significant results were found for the MDA-MB-231 colony graph (this is described on page 21), therefore significance bars are not shown on the graph..

  • Reviewer comment: Figure5: No bars

Author response: Thank you for your comment, the images in Figure 5 have been enlarged as much as possible to increase the size of the scale bars/resolution and still fit within the margins of the page.

  • Reviewer comment: Write tumor instead of tumour throughout the manuscript

Author response: We have chosen to use UK spelling. We can readily alter this at the typesetting stage if needed. This choice represents the geographical location of the study, and it is only fair for the manuscript to reflect that.

Round 2

Reviewer 2 Report

Thank You for the revision. The manuscript looks much improved . I would like to recommend it for publication in IJMS in its present form.